# Air-Conditioning Resource Management and Control Method based on Cloud Platform for Wind Power Consumption

**Kaixin Liang, Jinying Yu and Xin Wu ***

School of Electric and Electronic Engineering, North China Electric Power University, Beijing 102206, China

\* Correspondence: wuxin07@ncepu.edu.cn; Tel.: +86-186-1130-8185

**Abstract:** Air-conditionings have energy storage functions. Through reasonable aggregation control, the output tracking can be implemented for wind power with stronger fluctuation to enhance its utilization rate. Cloud technology and intelligent appliances enable the appliance vendor to implement information interaction with the air-conditioning through cloud platforms to realize flexible regulation. In this paper, a management and control method of air-conditioning based on cloud platform is established. Based on this structure, the air-conditionings are divided into several aggregation groups according to the similarity of parameters, and each group completes the consumption task collaboratively. The consumption evaluation model of the air-conditioning group is established. On this basis, the allocation problem on consumption task for the aggregated group is constructed to implement the optimal solution under the condition of guaranteeing the degree of completion and user comfort. Each group implements the control for air-conditioning inside the group through the sliding mode control model. The simulation experiment verifies that the algorithm can effectively follow the output of clean energy, while intervening less in the air-conditioning operation.

**Keywords:** air-conditioning management; cloud control; wind power tracking; air-conditioning grouping collaborative control

## 1. Introduction

Clean energy has the disadvantage of low utilization, which leads to a development restriction. Generally, clean energy including photovoltaic and wind power has the characteristics of intermittence and oscillation, and especially in situations with weak grids, it would be desirable to locate some ancillary services in the vicinity of the plant [1]. Several recent researches have dealt with the low utilization of clean energy in many ways, including demand side management, flywheel energy storage, and micro-grid. The presence of abundant load resources in power demand would enable it to provide ancillary services for the power supply side [2–7], for instance, the load following application in clean energy by controlling residential air-conditionings (ACs). As thermostatically controlled loads, ACs have the potential to store energy and account for a large proportion in power demand [1,8–12]. AC resources can be utilized to consume clean energy, especially to follow the detail of the oscillatory clean energy output because they have large capacities and can be controlled flexibly. It is significant to study consuming clean energy by modulating electrical demand of ACs [13].

In the demand side management, load resources in power demand can be regarded as an energy storage system that is capable of improving clean energy utilization. Researches in thermostatically controlled loads management are proposed in a lot of literature, especially in load control. The

aggregated model and controller are developed to modulate the electrical demand of aggregated loads in Callaway, Lu and Bashash's studies [1,9,10].

Lu has developed a state-queuing model of thermostatically controlled appliances to describe the dynamics of aggregated loads [8,9]. Based on Fokker-Planck, Callaway developed a system, which modulated aggregated power of ACs to deliver load following by adjusting the temperature set point offset [1]. Another method based on Callaway's study was developed in Bashash's study [10]. The dynamic of aggregated loads was described by the state-space model and the model was controlled by the Lyapunov controller. An improving stochastic control method was proposed, which had better robustness compared with a traditional stochastic controller [14]. The studies above consider AC resources controlled by one controller, and the methods showed good performance when AC resources were abundant, while they had degradation performance in the situation of ACs' diversity shortage. Some of the recent advances proposed the distributed control method including multiple controllers to reduce the negative effect caused by ACs' diversity shortage [11,15]. In Iacovella's study the thermostatically controlled loads were divided into major clusters. The control signal was broadcasted to all ACs in one cluster, which was a simple communication approach with obvious signal delay. In this paper, ACs fall into several groups and ACs in different groups subjected to independent signals.

At present, the existing researches on demand-side load management mainly focus on the load control algorithm. However, demand-side resources are physically dispersed, which is difficult for the implementation of centralized management. In order to make full use of demand-side load resources, a mechanism to enable the power grid to manage dispersed loads centrally should be constructed, and the communication between the power grid and the user side should be established. On the basis of the mechanism, the developed control algorithm of load resources can be carried out. However, there is still a lack of relevant discussion in load management foundation. With the development of smart home appliances and cloud technologies, home appliance vendors with cloud platforms can control their own loads centrally through the cloud platform, which provides the realization basis for loads centralized control. Additionally, the appliance vendors with cloud platforms can manage and control loads as aggregators. Based on the above, relevant research on load management and control has been done in this paper. We established load resource management and control structure with power grid, aggregator, and demand-side loads' participation.

To solve the above problems, an AC management and control method based on cloud platform is proposed in this paper. In this structure, appliance vendors with cloud platforms act as load aggregators to manage the scattered ACs. As the cloud platform contains ACs with different characteristics, aggregators need to divide the ACs into groups as the basic objects for collaborative control.

In what follows, the foundation of cloud control implementation is introduced in Section 2. Based on the foundation, an AC management and control method is developed in Section 3, which contains the AC group collaborative control process. Additionally, in Section 4, the performance of the proposed method is verified by tracking the wind power output.

## 2. Foundation of Cloud Control Implementation

AC is an important demand response resource that has the potential to provide the load following service for the power grid to consume clean energy. However, it is difficult to implement the centralized management of the scattered ACs. In order to realize the centralized control of ACs, it is important to acquire AC data in real time. Additionally, this problem is expected to be solved through the cloud platform. With the development of Internet of things, load data will be uploaded to the Internet. AC data can be obtained from the Internet and each AC will belong to at least one cloud platform.

In this paper, an AC management and control method based on cloud platform is proposed. Appliance vendors act as aggregators to manage and control the ACs through cloud platforms. Figure 1a shows the process of the AC cloud control method, including two parts: AC state data

acquisition and control instruction execution. The process of data acquisition is shown as route A: AC sends packets to routers via WIFI; the cloud platform binds the appliance terminal MAC (Media Access Control) address and the packets are transmitted to cloud platform through WAN (Wide Area Network); the cloud platform analyzes packets and stores the data in the database; and the power grid is connected to the cloud platform to obtain the AC state data and creates the control instruction according to the state data. The process of control instruction execution is shown as route B: The connection between the control terminal of power grid and cloud platform is established; the control signal is transmitted to the AC by cloud platform.

In Figure 1a, the control signal is calculated by the aggregated AC control model, which is established according to the state data of AC. There are different system parameters in the different AC groups. According to the system parameters, the ACs are divided into homogeneous aggregated groups as the basic objects of collaborative control.

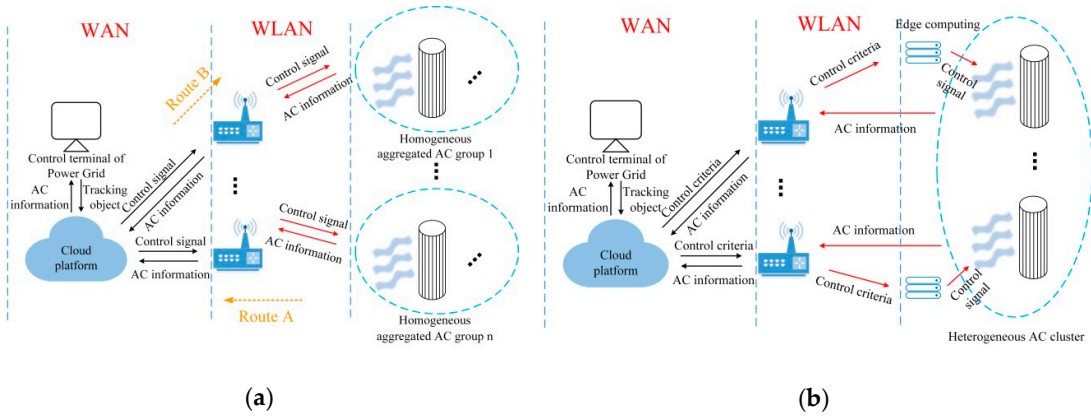

(**a**)  (**b**)

**Figure 1.** (**a**) AC (air-conditionings) group control structure based on cloud platform; (**b**) heterogeneous AC cluster control the structure based on cloud platform and edge computing.

In further research, centralized control and management for heterogeneous ACs are important. Figure 1b demonstrates the process of heterogeneous AC cloud control. The difference between (a) and (b) is that the edge computing is added to the demand side in the structure of AC cloud control. The power grid control terminal generates the AC control criterion containing multiple regulations for adjusting ACs in different states. After receiving the control criterion, according to the state information, the regulation of each AC is determined by the edge calculation module.

## 3. AC Management and Control Model Based on Cloud Platform

Based on the cloud control structure, an AC management and control model is proposed to track wind power output. Section 3.1 introduces a heat exchange model of single AC and establishes an aggregated AC model, which is used to estimate the aggregated power of the AC group. Section 3.2 presents the optimization selection of aggregated AC groups and task allocation strategy, and Section 3.3 describes the control system of AC groups. Finally, feasibility analysis of this method is introduced in Section 3.4.

### 3.1. Model of Single AC

Through the correlation between the internal temperature and switched-on/switched-off state of the AC system, the dynamics of the AC power is obtained, and the aggregated power consumption of an AC group is estimated.

The dynamic internal temperature evolution of an AC system is described by the first-order differential equation that contains the internal temperature $\theta_i(t)$ and the switched-on/switched-off state variable $s_i(t)$ of the AC:

$$\dot{\theta}_i(t) = \frac{1}{C_i R_i}(T_{a,i} - \theta_i(t) - s_i(t)R_i P_i), \, i = 1,2,\cdots,N_L,$$ (1)

where $T_{a,i}$ is the ambient temperature, $C_i[\text{kWh}/\,^\circ\text{C}]$ is heat capacity of AC $i$, $R_i[\,^\circ\text{C}/\text{kW}]$ is thermal resistance, and $P_i[\text{kW}]$ is the average power consumption. It is assumed that the ambient temperature remains constant for a period. The state variable $s_i(t)$ switches its value according to the switching criterion—as described in the following equation:

$$s_i(t) = \begin{cases} 0 & \text{if } s_i(t-\tau)=1 \,\&\, \theta_i(t) < \min\theta_i(t) \\ 1 & \text{if } s_i(t-\tau)=0 \,\&\, \theta_i(t) > \max\theta_i(t), \\ s_i(t-\tau) & \text{otherwise} \end{cases}$$ (2)

It is supposed that the ACs are in the refrigeration mode. When the internal temperature $\theta_i(t)$ exceeds the set temperature upper limit $\max\theta_i(t)$ and the AC is "OFF" in the previous period, the AC state at this period is switched to "ON". On the contrary, when the internal temperature $\theta_i(t)$ is below the temperature set point lower limit $\min\theta_i(t)$ and the AC is "ON" in the previous period, the AC state at this period is switched to "OFF". The limits $[\min\theta_i(t), \max\theta_i(t)]$ are related to the set temperature $T_{set,i}$ and the temperature deadband of AC $\Delta db, i$, shown as Equation (3).

$$\min\theta_i(t) = T_{set,i}(t) - \frac{\Delta db,i}{2}; \, \max\theta_i(t) = T_{set,i}(t) + \frac{\Delta db,i}{2},$$ (3)

According to the heat exchange model of a single AC system, cf. Equation (1)–(3), the estimated aggregated power consumption of AC group $j$ can be obtained. Supposing that there are $N_L^j$ ACs in group $j$ and the aggregated power consumption at time $t$ can be calculated:

$$P_A^j(t) = \frac{P^j}{\eta^j}\sum_{i=1}^{N_L^j} s_i^j(t), \, j = 1,2,\cdots,M,$$ (4)

where $\eta^j$ is the efficiency ratio of ACs in group $j$.

### 3.2. Grouping of AC Resources and Task Assignment for AC Groups

The existing aggregated AC control models mainly focus on the homogeneous AC control. In practice, if the specifications of the ACs are different, the corresponding equivalent thermal parameters and system parameters are also different. Therefore, AC resources need to be divided into homogeneous aggregated groups according to the parameters. In this paper, the ACs groups are managed as the basic objects. Additionally, from the single AC model, it can be seen that the AC system parameters include thermal resistance, heat capacity, set temperature, and average power, which affect the model together. Therefore, according to the four kinds of parameters, the load resources are divided into several groups that meet the requirements of the homogeneous aggregated model.

The selection and task assignment of aggregated AC groups should meet the following requirements:

Firstly, in order to ensure high tracking accuracy, the two norms of the difference between the total estimated power of the selected aggregated groups and the energy to be consumed are set as the objective function. The objective function is solved to select the AC groups suitable for participating in the task, as shown in Equation (5).

$$f = \min \sum_{t=0}^{T} \left\| P_{des} - \sum_{j \in \Theta} P_A^j \right\|, \tag{5}$$

where $\Theta = \{1, 2, \cdots, M\}$ is the set containing the number of $M$ groups of ACs. Additionally, $n$ groups are selected from the $M$ aggregated AC groups to participate in the load following service. $P_A^j$ represents the estimated power consumption of the aggregated AC group involved in the load following service and $P_{des}$ is the accommodation target.

Secondly, considering the impact on the users, the consumption task of the AC group is constrained within its adjustable range. Set $P_T^j(t)$ as the tracking task of aggregated AC group $j$ and it is adjusted within $[P_{A^-}^j(t), P_{A^+}^j(t)]$. The power consumption range $[P_{A^-}^j(t), P_{A^+}^j(t)]$ is related to the users' acceptable temperature range. It is supposed that the set temperature of group $j$ is adjusted within the users' acceptable range $[\min T_{acc}^j, \max T_{acc}^j]$. The relationship between power consumption range and temperature range is shown as:

$$P_{A^-}^j(t) = \frac{P^j}{\eta^j} \sum_{i=1}^{N_L^j} \min s_i^j(t), \, j = 1, 2, \cdots, M, \tag{6}$$

$$P_{A^+}^j(t) = \frac{P^j}{\eta^j} \sum_{i=1}^{N_L^j} \max s_i^j(t), \, j = 1, 2, \cdots, M, \tag{7}$$

where $\min s_i^j(t)$ and $\max s_i^j(t)$ are the switched-on/switched-off state variables with a set temperature of AC group $j$, calculated as the following Equations:

$$\min s_i^j(t) = \begin{cases} 0 & \text{if } \min s_i^j(t-\tau) = 1 \& \theta_i(t) < \min T_{acc}^j(t) - \dfrac{\Delta db^j}{2} \\ 1 & \text{if } \min s_i^j(t-\tau) = 0 \& \theta_i(t) > \min T_{acc}^j(t) + \dfrac{\Delta db^j}{2}, \\ \min s_i^j(t-\tau) & \text{otherwise} \end{cases} \tag{8}$$

$$\max s_i^j(t) = \begin{cases} 0 & \text{if } \max s_i^j(t-\tau) = 1 \& \theta_i(t) < \max T_{acc}^j(t) - \dfrac{\Delta db^j}{2} \\ 1 & \text{if } \max s_i^j(t-\tau) = 0 \& \theta_i(t) > \max T_{acc}^j(t) + \dfrac{\Delta db^j}{2}, \\ \max s_i^j(t-\tau) & \text{otherwise} \end{cases} \tag{9}$$

The $n$ AC groups that minimize the objective function can be obtained by intelligent search. The artificial immune algorithm is an excellent swarm intelligence search algorithm that can converge quickly and avoid local optimization. Additionally, it is used to solve the objective function in this paper.

The optimization process is as follows:

(1) Antigen recognition: It refers to the analysis process of the problem to be solved, and according to the analysis results, the appropriate objective function is constructed.
(2) Generation of initial antibody population: The feasible solution of the problem needs to be represented as an antibody in the solution space by coding. In general, the initial antibody population is randomly generated in the solution space, and each antibody is a real vector. Supposing that there are a total of $M$ groups of aggregated ACs, $n$ groups are selected to participate in the accommodation. The antibody is the number combination of the $n$ AC groups.

(3) Update individuals: The individuals are updated by executing immune operation, which contains the clone, crossover, and mutation.

(4) Evaluation of individuals: Individuals are evaluated through calculating affinity, concentration, and reproductive rate. Individual affinity contains antibody–antigen affinity and antibody–antibody affinity. The affinity calculation equations are shown as follows:

$$A_v = \frac{1}{f}, \tag{10}$$

$$S_{v,s} = \frac{k_{v,s}}{n}, \tag{11}$$

where *f* is the objective function value cf. Equation (5), $A_v$ is the affinity of antibody *v*, and $S_{v,s}$ is the affinity between antibody *v* and *s*. $k_{v,s}$ is the similarity between antibody *v* and *s* and *n* is the antibody length.

(5) Parent individuals: The antibody was screened by immune balance operation and immune selection operation, and the parental antibody was formed.

Immune balance is a mechanism to ensure population diversity. To regulate the concentration of individuals, the balance operation containing the promoting operator and inhibiting operator should be executed. The individuals with excessive concentration should be inhibited, which can effectively prevent the problem of algorithm precocious due to excessive concentration. According to the affinity and concentration of individuals, the reproductive rate of the individuals is calculated:

$$C_v = \frac{1}{N}\sum_{i \in N}S_{v,s}; \tag{12}$$

$$P_r = \alpha \frac{A_v}{\sum A_v} + (1-\alpha)\frac{C_v}{\sum C_v}, \tag{13}$$

where $C_v$ is the concentration of antibody *v*; *N* is the number of iterations; $P_r$ is the reproductive rate, and $\alpha$ is the diversity evaluation parameter of antibody *v*.

(6) Record optimal individuals.

(7) Judging whether the iteration meets the end condition, if it meets the end condition, the recorded optimal individuals is the optimal solution of the objective function, if not, the antibody population should be updated by crossover and mutation and the above process should be repeated until the optimal solution is obtained.

(8) Judge whether the maximum iteration *N* has been achieved. If the end request is met, then output the optimum individual or go back to Equation (3).

The flow of artificial immune algorithm is shown in Figure 2.

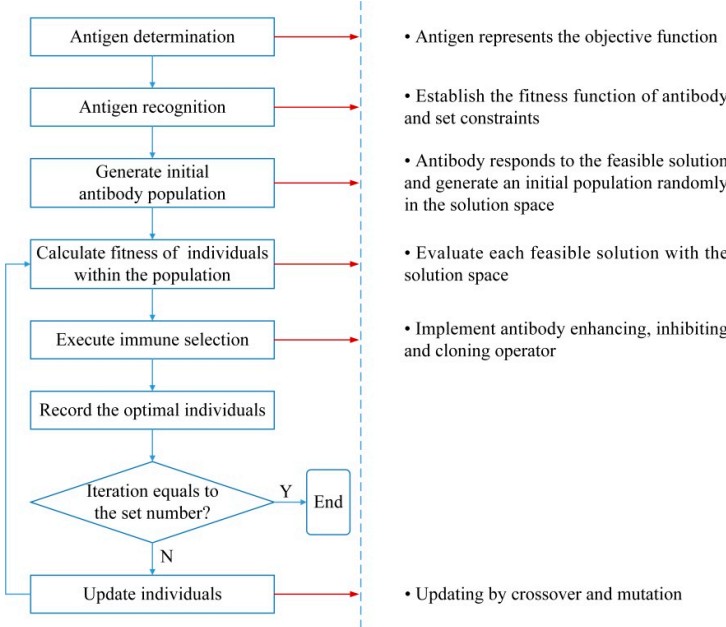

**Figure 2.** The AC groups optimization process by artificial immune algorithm.

Through the immune algorithm, the optimal solution is obtained. The estimated power consumption of selected AC group $j$ is denoted as $P_{A*}^{j}(t)$, and it is used to assign the tracking tasks for $n$ selected AC groups, shown as follows:

$$P_T^j(t) = \begin{cases} \Delta^j(t) + P_{A*}^j(t), & 0 < \Delta^j(t) < P_{A*+}^j(t) - P_{A*}^j(t) \ or \ 0 > \Delta^j(t) > P_{A*-}^j(t) - P_{A*}^j(t) \\ P_{A*+}^j(t), & \Delta^j(t) \geq P_{A*+}^j(t) - P_{A*}^j(t) \\ P_{A*-}^j(t), & \Delta^j(t) \leq P_{A*-}^j(t) - P_{A*}^j(t) \end{cases}, \qquad (14)$$

where $P_T^j(t)$ is the task of group $j$ at time $t$, adjusted within the adjustable range of the AC group and $\Delta^j(t)$ is the increment of the task assigned to a group $j$:

$$\Delta^j(t) = \frac{P_{A*}^j(t)}{\sum\limits_{j \in \Theta} P_{A*}^j}(P_{des} - \sum\limits_{j \in \Theta} P_{A*}^j), \qquad (15)$$

### 3.3. Modeling of AC Group and Collaborative Control

Equations (1)–(4) can describe the aggregated ACs dynamics effectively, while it is difficult to design the control module due to the characteristics of the multiple inputs single output. Hence, a single input single output model is required. Bashash et al. proposed a state–space model to describe the dynamics of aggregated ACs and it is a single input single output system. Figure 3 illustrates a finite-difference discretization of the state–space model at time $t$ and the state–space is divided into $Q$ segments. The loads distributed on the temperature axis, $\alpha_{on}$ and $\alpha_{off}$ represent the transport rates of loads at ON and OFF states, calculated by:

$$\alpha_{on}(T_a, T) \cong \bar{\alpha}_{on}(T_a, T_{set}) = \frac{1}{CR}(T_a - T_{set} - RP), \qquad (16)$$

$$\alpha_{off}(T_a, T) \cong \bar{\alpha}_{off}(T_a, T_{set0}) = \frac{1}{CR}(T_a - T_{set0}),\tag{17}$$

where, $T_{set0}$ is the initial temperature set point. Neglecting the temperature offset around $T_{set0}$, the load transport rates are approximated to be average load transport rates $\bar{\alpha}_{on}$, $\bar{\alpha}_{off}$.

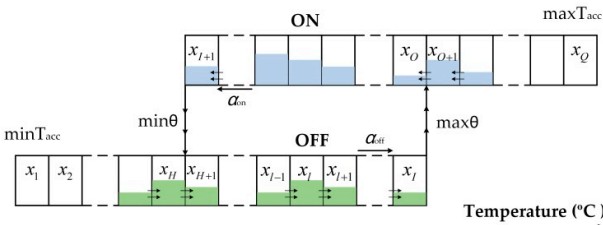

**Figure 3.** Finite-difference discretization of the system for state-space model derivation.

The state-space model is described in Bashash's study [10] and we do not repeat it. The aggregated power consumption at time *t* is calculated by Equation (18):

$$P_{Agg}(t) = \frac{P}{\eta} \sum_{j=I+1}^{Q} x_j,\tag{18}$$

where $x_j$ is the number of ACs in segment *j*.

The state–space model can be written in the form of the bilinear equation, as shown in Equation (19). It is a standard form in the area of automatic control.

$$\dot{x}(t) = Ax(t) + Bx(t)u(t), u(t) = \dot{T}_{set}(t)$$
$$y(t) = Cx(t)\tag{19}$$

where $x(t) = [x_1(t), x_2(t), \cdots, x_Q(t)]^T$ is the state vector and $y(t) = P_{Agg}(t)$ is the output of Equation (18). $A(\bar{\alpha}_{on}, \bar{\alpha}_{off})$ is the coefficient matrix structured as Equation (20) and the other coefficient matrixes $B = A(-1, -1)$, $C = [0, \ldots, 0|_I, P/\eta, \ldots, P/\eta|_Q]$.

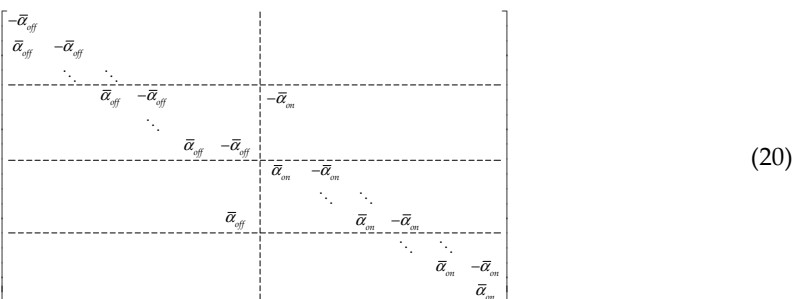

To ensure that the aggregated AC groups complete the tracking task, a control algorithm that matches the aggregated AC model is introduced below. In reality, the wind power output is fluctuates because of the uncertainties in the environment. Therefore, we need a dynamic control algorithm with the characteristics of fast response and flexible control to deal with the problem. The sliding mode model has the advantage of insensitivity to disturbance and fast response. It can be carried by cloud platform as the aggregated AC group control algorithm [16,17].

Refer to the study of Bashash [10], the sliding mode function is defined as:

$$s = e(t) = P_{des}(t) - P_{Agg}(t),\tag{21}$$

Then, the Lyapunov candidate function is defined as:

$$V(t) = \frac{1}{2}s^2 , \tag{22}$$

Treating the entire thermostatic ON state as a bulk control volume, the rate of change of aggregated AC group's power is then given by:

$$\dot{P}_{Agg} = \frac{P}{\eta}\dot{N}_{on}(t) , \tag{23}$$

where $\dot{N}_{on}(t)$ is the total number of loads over the ON state. The rate at which $\dot{N}_{on}(t)$ varies is governed by the difference between the boundary fluxes entering and leaving the ON state as follows:

$$\dot{N}_{on}(t) = [\alpha_{off}(t, \max T_{acc}) - \dot{T}_{set}]X_{off}(t, \max T_{acc}) + [\alpha_{on}(t, \min T_{acc}) - \dot{T}_{set}]X_{on}(t, \min T_{acc}) , \tag{24}$$

where $\alpha_{off}(t, \max T_{acc})$ and $\alpha_{on}(t, \min T_{acc})$ represents the instantaneous local average transport rates at the deadband boundaries. Note that these two quantities have opposite signs, and are time-varying in general, due to load heterogeneity and the variation of ambient and set point temperatures. Let

$$f(\alpha, X) = \frac{P}{\eta}(\alpha_{off}(t, \max T_{acc})X_{off}(t, \max T_{acc}) + \alpha_{on}(t, \min T_{acc})X_{on}(t, \min T_{acc})) , \tag{25}$$

$$g(X) = \frac{P}{\eta}(X_{off}(t, \max T_{acc}) + X_{on}(t, \min T_{acc})) \geq 0 , \tag{26}$$

$$u(t) = \dot{T}_{set}(t) , \tag{27}$$

$$\dot{V}(t) = e(t)\left\{\dot{P}_{des}(t) - f(\alpha, X) + g(X)u(t)\right\} , \tag{28}$$

To ensure the stability of control system $\dot{V}(t)$ must be negative for all nonzero $e(t)$ [10]. Additionally, to satisfy the requirement, the control input $u(t)$ is structured as:

$$\begin{cases} u(t) = -\kappa(t)\text{sgn}\left\{\dfrac{e(t)}{\varepsilon}\right\} \\ \kappa(t) > \left|\dfrac{P_{des}(t) - f(\alpha, x)}{g(x)}\right|' \end{cases} \tag{29}$$

The final control law for the set temperature is obtained by integrating Equation (29) as follows:

$$\Delta T_{set}(t) = -\int_0^t \kappa(\tau)\text{sgn}(e(\tau))d\tau , \tag{30}$$

*3.4. The Feasibility of Data Acquisition and the Influence of Communication Delay on the System*

In this paper, an AC cloud management and control method is proposed, and the scattered ACs can be centralized controlled via the cloud platform. As the algorithm has the requirement of data transmission speed in practical application, the real-time data acquisition of the AC system and fast control signal transmission must be guaranteed. Here, the feasibility of data acquisition is analyzed and the delay of data transmission is tested.

With the development of Internet of things, AC data can be obtained from the Internet and each AC will belong to at least one cloud platform. Taking the remote control of intelligent home appliances as an example, the intelligent home appliances and cloud platform is connected via the Internet. Users can remotely control the appliances by APP (Application). Additionally, it demonstrates that the cloud platform can control loads directly. There are lots of loads under the

cloud platform, thus cloud platforms participating in loads management will provide the foundation for the implementation of the load control technology. Therefore, in this paper, the system parameters such as the state of AC system, the set temperature, the average power, the heat capacity, and the thermal resistance can be obtained through the cloud platform.

With the breakthrough of mobile network technology, the speed of communication speed has been improved. Additionally, the real-time remote load control can be guaranteed. According to the method proposed in this paper, the real-time control of ACs is the basis to ensure tracking accuracy. In order to test the delay of data transmission over WAN, we measure the communication delay of remote control for Midea intelligent AC. As the geographical position influences the signal intensity of the mobile network, we tested the delay many times in many places of different signal intensity. The results show that the control signal has a maximum delay of 2 s when the signal is weak, and under normal conditions, the signal delay is less than 0.5 s. If the cloud platform directly controls appliances, the communication between a mobile phone and cloud platform will be ignored, and the data transmission delay will be less than two seconds, which has no obvious effect on the control process and it can be ignored.

## 4. Simulation and Analysis

To verify the performance of the collaborative group control method, an experiment is designed to evaluate the method in terms of the tracking accuracy and influence on users. In this experiment, there are 4000 ACs belonging to five cloud platforms participating in the tracking service.

### 4.1. Grouping of AC Resources

In this paper, the ACs are controlled by the cloud platform. The average power, thermal resistance, heat capacity, and initial set temperature of the AC system are obtained by the cloud platform. The initial set temperature and average power can be obtained directly, while the thermal resistance and heat capacity are related to the thermal conductivity, structure, and orientation of the building. Therefore, the thermal resistance and heat capacity should be calculated according to known parameters collected by the cloud platform. The process is as follows:

Assuming that the air conditioning is in the refrigeration mode, the ambient temperature and average power are known. In the ON state, the internal temperature of the AC system at $r$ and $r + 1$ time is collected and denoted as $T_i(r)$, $T_i(r+1)$; In the OFF state, the internal temperature of air conditioning system at $k$ and $k + 1$ time is collected and denoted as $T_i(k)$, $T_i(k+1)$. By merging the above known variables into the difference equation derived from Equation (1), we can obtain Equations (31)–(32), shown as follows:

$$T_i(k+1) = T_{a,i}(k+1) - \left(T_{a,i}(k+1) - T_i(k)\right)e^{\frac{-1}{R_iC_i}}, \tag{31}$$

$$T_i(r+1) = T_{a,i}(r+1) + P_iR_i - \left(T_{a,i}(r+1) + P_iR_i - T_i(r)\right)e^{\frac{-1}{R_iC_i}}, \tag{32}$$

Solving Equations (31) and (32), the thermal resistance and heat capacity of AC system are expressed as Equations (33)–(34):

$$R_i = \frac{\left(T_{a,i}(r+1) - T_i(r)\right)\left(T_{a,i}(k+1) - T_i(k+1)\right)}{\left(T_{a,i}(r+1) - T_i(r+1)\right)\left(T_{a,i}(k+1) - T_i(k)\right)}, \tag{33}$$

$$C_i = \frac{-P_i\left(T_{a,i}(r+1) - T_i(r+1)\right)\left(T_{a,i}(k+1) - T_i(k)\right)}{\ln\left(\frac{T_{a,i}(k+1) - T_i(k+1)}{T_{a,i}(k+1) - T_i(k)}\right)\left(T_{a,i}(r+1) - T_i(r)\right)\left(T_{a,i}(k+1) - T_i(k+1)\right)}, \tag{34}$$

Finally, the 4000 ACs are divided into 70 homogeneous aggregated groups and the cluster centers of the aggregated AC groups are shown in Table 1:

**Table 1.** Cluster center of the aggregated AC groups.

| Parameter | Value |
|---|---|
| $P$ kW | 14; 11.2; 8.4; 5.6; 2.8 |
| $R$ °C/kW | 2; 2.5; 3.33; 5; 10 |
| $C$ kWh/°C | 10; 8; 6; 4; 2 |
| $T_{ser0}$ °C | 20; 21; 22; 23; 24; 25; 26 |

### 4.2. Consumption Task Assignment for AC Groups

The immune algorithm is utilized to select the aggregated AC group involved in the tracking task, and finally 700 ACs of 10 groups are selected to track the wind power output. In Figure 4, it can be seen that the estimated total power is similar to the tracking target.

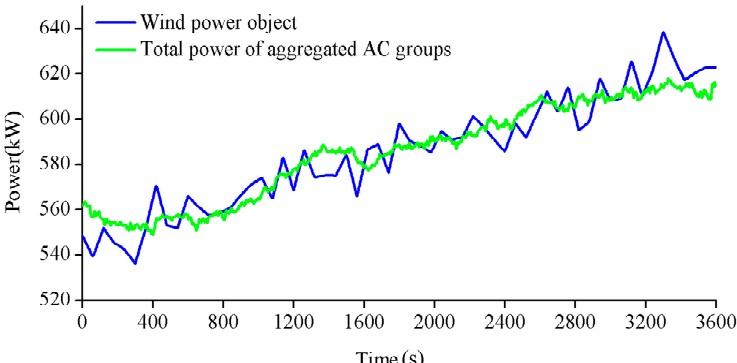

**Figure 4.** Wind power tracking target and estimated total power of aggregated AC groups.

Additionally, the parameters of the AC groups involving tracking service are shown in Table 2.

**Table 2.** Parameters of the selected AC groups.

| No. | $R$ °C/kW | $C$ kWh/°C | $P$ kW | $T_{ser0}$ °C | $NL$ |
|---|---|---|---|---|---|
| A3 | 2.5 | 8 | 11.2 | 24 | 82 |
| A6 | 5 | 4 | 5.6 | 24 | 53 |
| B3 | 2 | 10 | 14 | 20 | 60 |
| B6 | 3.33 | 6 | 8.4 | 20 | 115 |
| C11 | 5 | 4 | 5.6 | 20 | 85 |
| C12 | 5 | 4 | 5.6 | 26 | 50 |
| D13 | 2 | 10 | 14 | 26 | 60 |
| D14 | 10 | 2 | 2.8 | 23 | 66 |
| E4 | 2 | 10 | 14 | 23 | 85 |
| E7 | 2 | 10 | 14 | 26 | 44 |

After determining the AC groups involved in the tracking service, tracking tasks are assigned to the selected AC groups, as shown in Figure 5. Figure 5a is the estimated power trajectory of AC groups; Figure 5b is the assigned task. The tracking target of the aggregated AC group and the assigned task is within the adjustable range of the AC group. Additionally, the less interventions carried out in the AC control process, the less impact there is on the users.

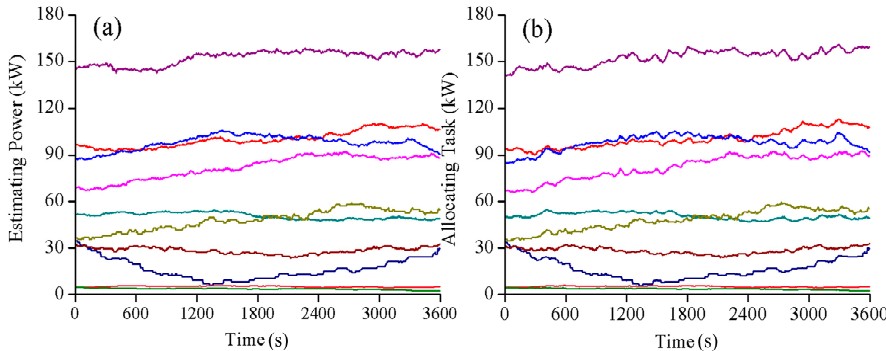

**Figure 5.** (**a**) The estimated power trajectory of AC groups; (**b**) assigned task of selected AC groups.

### 4.3. Algorithm Performance Analysis

To validate the performance of the collaborative group control method, the AC groups are controlled to track the wind power object in Figure 4. Additionally, Bashash's algorithm [10] is compared with the algorithm in this paper. In Bashash's study [10], a discrete finite difference state space model for homogeneous aggregated ACs is established, and a sliding mode control system is established for the model. Based on the model proposed by Bashash [10], a collaborative grouping method of ACs is proposed in this paper. According to the estimated power of the AC groups, the optimal tracking tasks are assigned to AC groups. Additionally, we think that the accuracy of tracking control can be improved and the impact on users can be reduced when the tracking task is close to the estimated power of the AC group. In order to verify this conclusion, we establish a comparison experiment and the results are as follows:

It is assumed that the two methods control the same ACs and the tracking targets are the same. The difference between the two methods is that ACs are the group control and optimal tracking tasks are assigned to the AC groups in our method.

Figure 6a is the tracking process of the two methods. It can be seen that both the traditional method and the proposed method can accurately track the target. Additionally, in Figure 6b,c, the tracking error and error distribution of the two methods are shown. In terms of tracking accuracy, the results show that the collaborative group control method has a better performance than the comparison method.

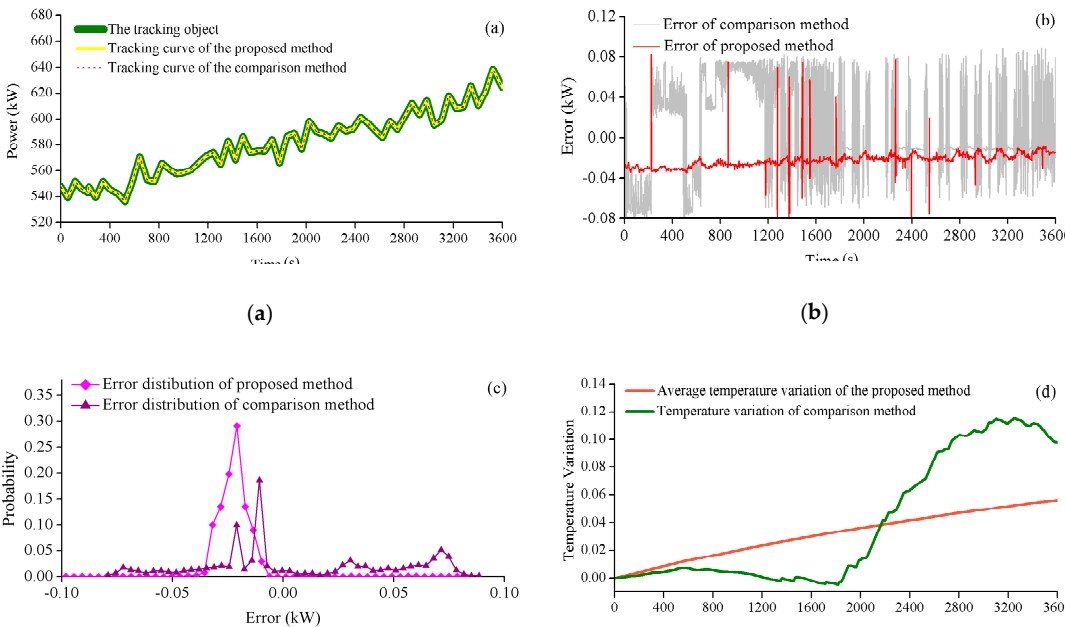

(**c**)                        (**d**)

**Figure 6.** (**a**) Wind power tracking of two methods; (**b**) tracking errors of proposed method and comparison method; (**c**) error distribution of proposed method and comparison method; and (**d**) set temperature variation of the proposed method and comparison method.

The effects of the two methods on user comfort can be evaluated by comparing the set temperature changes of AC in the control process. Figure 6d is the set temperature variation comparison of the two methods in the control process of air-conditioning. The variation of collaborative control method in Figure 6d is equal to the average of the 10 AC groups set temperature variation. Obviously, the variation of the proposed method is less than the traditional method and it means that our method has good performance both in terms of tracking accuracy and its effects on users.

## 5. Conclusions

In this paper, an AC control and management method based on cloud platform for wind power consumption is established. The scattered AC resources are aggregated into controllable groups through the cloud platform, which is convenient for centralized management and provides ancillary services for the power grid. According to the system parameters, the AC resources are divided into several homogeneous aggregated groups, and the aggregated power of the AC group is estimated. Combined with the AC information fed back by the cloud platform and the wind power target to be tracked, the power grid optimally selects the aggregated AC groups and assigns a tracking task to each AC group. Additionally, the cloud platform collaboratively controls the aggregated AC groups to complete the tracking task. In this paper, according to the users' requirements and wind power output to be tracked, the optimal task allocation algorithm is established. Firstly, according to the estimated power and tracking target of the AC groups, the aggregated AC group participating in the task is selected optimally. Secondly, combined with the adjustable range of each AC group, the tracking task is assigned to each group. Finally, the sliding mode control model is used to control the aggregated AC group to complete the tracking task.

The simulation results demonstrate that the proposed method can follow the objective wind power effectively and meet users' requirements. Compared with the baseline control method, the AC set temperature varies within a smaller range and the method of this paper has a high tracking accuracy, which shows that the proposed method has a lower impact on users. In addition, due to the group management of AC resources, the proposed method is more flexible in tracking the output.

The managing framework based on the cloud platform can regulate the entire load under the cloud. Being similar to air-conditioning, electric water heaters also have the potential to be controllable load. Further research will focus on developing the integrated control model of electric water heater and air-conditioning to balance power supply and demand in the integrated energy system.

**Author Contributions:** X.W. and K.L. conceived and designed the experiments; K.L. performed the simulation; K.L. and J.Y. analyzed the results; and K.L. and J.Y. wrote and revised the paper.

**Funding:** This research was funded by [the Fundamental Research Funds for the Central Universities of China] grant number [2018MS001] and [Natural Science Foundation of Beijing Municipality under grant] grant number [3172034].

**Conflicts of Interest:** The authors declare no conflict of interest.

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
