# Peer review of "Air-Conditioning Resource Management and Control Method based on Cloud Platform for Wind Power Consumption"

_processes, doi:10.3390/pr7070467_

Round 1

Reviewer 1 Report

Really interesting paper, I think it could be improved making some clustering pre-processing using k-means algorithms or even testing proposed results against a k-means algorithm.

Reviewer 2 Report

This paper presents a new way of managing smart devices, focusing on AC units, with the purpose of load balancing a renewable energy power grid. The authors have applied a couple of novel techniques such as the usage of artificial immune system algorithm for optimization. 

Due to the lack of a real use case the authors are basing their evaluation on a simulation, based on data generated by them, which is not an ideal approach for solidifying their results. Also, the baseline methodology is not clearly described, which reduces the scientific soundness of this work even more.

The language of this paper is heavily flawed, it needs a major review by a qualified English speaker. It contains typos, erroneous wording and grammar errors which prohibits the reader from focusing on the content. At some points it is not clear what the authors are trying to say.

Equations 5 and 6 need a more clear explanation. Also, the term "prediction power" in line 175 is not defined so we do not know what it refers to, is it the predicted power consumption, or perhaps the prediction confidence ?

In lines 195-197 the authors are explaining equation 7 but they are also using symbols from equation 8, which is not really one function but three. These equations need to be separated and more clearly referenced.

In line 227 the authors reference Equation 10 but the mentioned output is more fitting to equation 13, possibly a typo.

Equation 15 is actually a matrix, which should be referenced as a Table or a Figure, not an Equation.

Equation 18 has no reference in the text and no clear explanation.

Round 2

Reviewer 2 Report

Most of the comments have been successfuly resolved, providing a much more clear presentation of the research methodology and results. 

The language of the paper still needs improvement although significant steps have been taken. Now at all points of the paper the reader can understand what the authors want to say but there are still some vocabularly and grammar faults that should be addressed in order to improve the image of the paper.

Finally, the baseline methodology (the one that the novel methodology is compared to, refered to as "the traditional method") is still not clearly defined, is the AC grouping the only difference ?  
